# Global Gated Mixture of Second-order Pooling for Improving Deep Convolutional Neural Networks

**Qilong Wang**[1,2,*,†]**, Zilin Gao**[2,*]**, Jiangtao Xie**[2]**, Wangmeng Zuo**[3]**, Peihua Li**[2,‡]

[1]Tianjin University, [2]Dalian University of Technology, [3] Harbin Institute of Technology
`qlwang@tju.edu.cn, gzl@mail.dlut.edu.cn, jiangtaoxie@mail.dlut.edu.cn`
`wmzuo@hit.edu.cn, peihuali@dlut.edu.cn`

## Abstract

In most of existing deep convolutional neural networks (CNNs) for classification, global average (first-order) pooling (GAP) has become a standard module to summarize activations of the last convolution layer as final representation for prediction. Recent researches show integration of higher-order pooling (HOP) methods clearly improves performance of deep CNNs. However, both GAP and existing HOP methods assume unimodal distributions, which cannot fully capture statistics of convolutional activations, limiting representation ability of deep CNNs, especially for samples with complex contents. To overcome the above limitation, this paper proposes a global Gated Mixture of Second-order Pooling (GM-SOP) method to further improve representation ability of deep CNNs. To this end, we introduce a sparsity-constrained gating mechanism and propose a novel parametric SOP as component of mixture model. Given a bank of SOP candidates, our method can adaptively choose Top-$K(K > 1)$ candidates for each input sample through the sparsity-constrained gating module, and performs weighted sum of outputs of $K$ selected candidates as representation of the sample. The proposed GM-SOP can flexibly accommodate a large number of personalized SOP candidates in an efficient way, leading to richer representations. The deep networks with our GM-SOP can be end-to-end trained, having potential to characterize complex, multi-modal distributions. The proposed method is evaluated on two large scale image benchmarks (i.e., downsampled ImageNet-1K and Places365), and experimental results show our GM-SOP is superior to its counterparts and achieves very competitive performance. The source code will be available at `http://www.peihuali.org/GM-SOP`.

## 1 Introduction

Deep convolutional neural networks (CNNs) have achieved great success in a variety of computer vision tasks, especially image classification [25]. During the past years, deep CNN architectures have been widely studied and achieved remarkable progress [34, 36, 13, 17]. As one standard module in deep CNN architectures [36, 13, 17, 5, 16], global average pooling (GAP) summarizes activations of the last convolution layer for final prediction. However, GAP only collects first-order statistics while neglecting richer higher-order ones, suffering from limited representation ability [7]. Recently, some researchers propose to integrate trainable higher-order pooling (HOP) methods (e.g., second-order and third-order pooling) into deep CNNs [19, 29, 39, 27, 8], which distinctly improve representation ability of deep CNNs. However, both GAP and existing HOP methods adopt the unimodal distribution

---

[*]The first two authors contribute equally to this work.
[†]This work was mainly done when he was with Dalian University of Technology.
[‡]Peihua Li is the corresponding author.

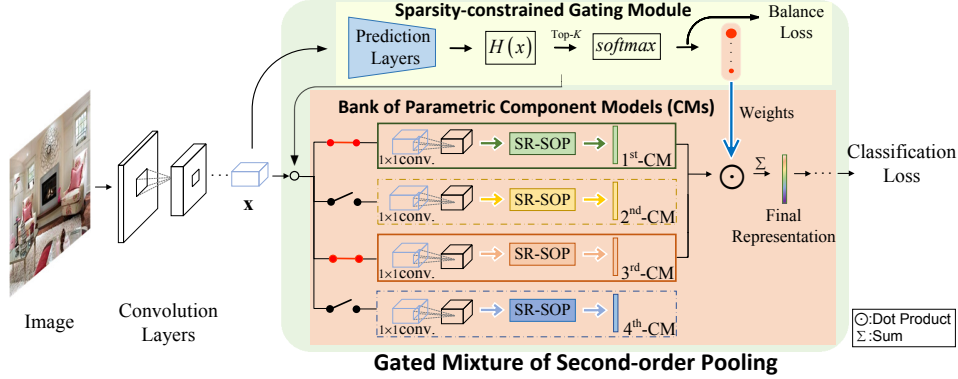

Figure 1: Overview of deep CNNs with the proposed global Gated Mixture of Second-order Pooling (GM-SOP). The sparsity-constrained gating module adaptively selects Top-$K$ parametric SR-SOP (indicated by solid rectangles) from a bank of $N$ candidate component models given a sample $\mathbf{X}$, and the final representation is generated by weighted sum of outputs of $K$ selected CMs. For brevity here we take $N = 4$, $K = 2$ as an example.

assumption to collect statistics of convolutional activations. As illustrated in Figure 1, input images often contain multiple objects or parts, leading their distributions of convolutional activations usually are very complex (e.g., mixture of multiple unimodal models). As such, unimodal distributions cannot fully capture statistics of convolutional activations, which will limit performance of deep CNNs.

One natural idea to overcome the above limitation is ensemble of multiple models for summarizing convolutional activations. However, direct ensemble of all component models (CMs) in mixture model will suffer from very high computational cost as number of CMs gets large, while a small number of CMs may be insufficient for characterizing complex distributions. Moreover, simple direct ensemble will make all CMs tend to learn similar characteristics, since they receive identical training samples. These factors heavily limit the representation ability of mixture model (refer to the results in Section 3.2). Inspired by recent work [33], we propose an idea of gated mixture model to solve the above issues. Our gated mixture model is composed of a sparsity-constrained gating module and a bank of candidate CMs. Given an input sample, the sparsity-constrained gating module adaptively selects Top-$K$ CMs from $N(N \gg K)$ candidates according to assigned weights, and then weighted sum of outputs of $K$ selected CMs is used to generate final representation. In this way, our gated mixture model can accommodate efficiently a large number of CMs because only $K$ ones are trained and used to generate representation given a sample. Furthermore, different CMs will receive different training samples so that they can capture personalized characteristics of convolutional activations during training. As suggested in [33], we employ an extra balance loss to eliminate self-reinforcing phenomenon, guaranteeing as many candidate CMs as possible be adequately trained.

The CM plays a key role in gated mixture model. Compared with first-order GAP, HOP can capture more statistical information, achieving remarkable improvement in either shallow models [4, 23] or deep architectures [19, 29, 8, 27]. As shown in [26], the comparisons on both large-scale image classification [9] and fine-grained visual recognition demonstrate matrix square-root normalized second-order pooling (SR-SOP) outperforms other HOP methods and achieves promising performance in deep architectures. In view of effectiveness of SR-SOP, it seems to be a good choice for CM. However, there exist two problems lying in usage of SR-SOP. Firstly, SR-SOP is a parameter-free model, which cannot be individually learned. Meanwhile, SR-SOP assumes data distribution obeys a Gaussian, which may not always hold true. To address these problems, this paper proposes a parametric SR-SOP method[4], which enables candidate CMs to be individually trained with negligible additional cost. Besides, underlying the parametric SR-SOP is estimation of covariance in the generalized Gaussian setting which has better modeling capability than SR-SOP. Based on the parametric SR-SOP, we propose a global Gated Mixture of Second-order Pooling (GM-SOP),

which is illustrated in Figure 1. Our GM-SOP can effectively exploit a large bank of personalized SOP models to generate more discriminative representations.

We evaluate the proposed GM-SOP method on two large scale image benchmarks, i.e., downsampled ImageNet-1K [9] and Places365 [44] that are introduced by [6] and this paper, respectively. As described in [6], downsampled ImageNet is a promising alternative to CIFAR10/100 datasets, as it is large-scale and more challenging, which postpones the saturation risk on CIFAR (as observed in `http://karpathy.github.io/2011/04/27/manually-classifying-cifar10/`). Compared with standard-size ImageNet, downsampled ImageNet allows much faster experiments and lower computation requirement while maintaining similar characteristics with respect to analysis of networks[6]. The contributions of our paper are three-fold. (1) We, for the first time, introduce a global gated mixture of pooling model into prevalent deep CNN architectures. This goes beyond the existing global average/covariance (second-order) pooling, possessing the potential to capture complex, multi-modal distributions of convolutional activations. (2) We propose parametric second-order models as essential components of mixture model. These components can be trained individually for modeling richer feature characteristics than simple second-order pooling. (3) We perform extensive experiments on two large-scale benchmarks for evaluating and validating the proposed methods, which have proven to achieve much better results than the counterparts.

## 2   Gated Mixture of Second-order Pooling (GM-SOP)

In this section, we introduce the proposed Gated Mixture of Second-order Pooling (GM-SOP) method. We first describe a general idea of gated mixture model, and then propose a parametric matrix square-root normalized second-order pooling (SR-SOP) method as our component model. Finally, the GM-SOP is integrated into deep CNNs in an end-to-end learning manner.

### 2.1   Gated Mixture Model

**Mixture Model**   The mixture model (e.g., finite mixture distributions [12, 30] or mixture of experts [21, 22]) is widely used to characterize complex data distribution or improve discrimination ability of supervised system through ensemble of multiple component models (CMs). In general, mixture model can be formulated as the weighted sum of all CMs, i.e.,

$$\mathbf{y} = \sum_{i=1}^{N} \omega_i(\mathbf{X}) M_i(\mathbf{X}), \ s.t., \ \sum_{i=1}^{N} \omega_i(\mathbf{X}) = 1, \tag{1}$$

where $N$ is the number of CMs. $\omega_i(\mathbf{X})$ and $M_i(\mathbf{X})$ indicate weight and output of $i$-$th$ CM given input $\mathbf{X}$, respectively. The mixture model in Eq. (1) consists of a weight (probability) function and a set of parametric CMs. Given specific forms of weight function and parametric CMs, mixture model in Eq. (1) can be learned by using gradient learning algorithm [21] or Expectation-Maximization (EM) algorithm [22, 30].

**Sparsity-constrained Gating Module**   Given an input sample, contribution of each CM in mixture model is decided by the corresponding weight through either computation of posterior probability in mixture distributions [12, 30] or a gating network in mixture of experts [21, 22]. However, they both acquiesce to allow every input sample to participate in training of all CMs. It will suffer from high computational cost when number of CMs is large. Meanwhile, CMs with small weights may bring noise into final representation [41]. Inspired by [33], we exploit a sparsity-constrained gating module as the weight function to overcome the above issues, where weights are learned by explicitly imposing a sparse constraint. As illustrated in Figure 1, we first pass $\mathbf{X}$ throughout a group of prediction layers with parameters $\theta_g$, i.e., $f(\theta_g; \mathbf{X})$. Then, weights are outputted by using a fully-connected layer with additional noise perturbations, i.e.,

$$H_i(f(\theta_g; \mathbf{X})) = \mathbf{W}_i^g f(\theta_g; \mathbf{X}) + \gamma \cdot \log(1 + \exp(\mathbf{W}_i^n f(\theta_g; \mathbf{X}))). \tag{2}$$

Here, $\mathbf{W}_i^g$ and $\mathbf{W}_i^n$ are $i$-$th$ row of parameters of fully-connected layer and additional noise, respectively. $\gamma$ is a random variable sampled from a normal distribution. To make the learned weights $H(f(\theta_g; \mathbf{X}))$ be sparse, only the $K$ largest weights are kept and remaining ones are set to be negative infinity, denoted as Top-$K(H_i(f(\theta_g; \mathbf{X})))$. Finally, a softmax function is used to normalize the

weights. To sum up, the weight function can be written as

$$\omega_i(\mathbf{X}) = \frac{\exp(\text{Top-}K(H_i(f(\theta_g; \mathbf{X}))))}{\sum_{i=1}^{N} \exp(\text{Top-}K(H_i(f(\theta_g; \mathbf{X}))))}. \tag{3}$$

**Balance Loss**    The sparsity-constrained gating module makes each sample participate in training of $K$ CMs. However, as shown in [33] and the results of Section 3.2, such gating module has a self-reinforcing phenomenon (i.e., only the same few CMs receive almost all samples while remaining ones have rarely been trained), decreasing representation ability of mixture model. As suggested in [33], we introduce an extra balance loss which is a function of weights defined as follows:

$$L_B = \alpha \left( \frac{std(\sum_{s=1}^{S} \boldsymbol{\omega}(\mathbf{X}_s))}{\mu(\sum_{s=1}^{S} \boldsymbol{\omega}(\mathbf{X}_s))} \right)^2, \tag{4}$$

where $\mathbf{X}_s$ is $s$-th training sample in a mini-batch of $S$ samples and $\boldsymbol{\omega}(\mathbf{X}_j) = [\omega_1(\mathbf{X}_j), \dots, \omega_N(\mathbf{X}_j)]$ is the weight function in Eq. (3); $std(\mathbf{v})$ and $\mu(\mathbf{v})$ denote standard deviation and mean of vector $\mathbf{v}$, respectively; $\alpha$ is a tunable parameter. The loss $L_B$ is to constrain that all CMs are adequately trained.

## 2.2   Component Model of GM-SOP

Besides the weight function, component model (CM) plays an indispensable role in gated mixture model. Motivated by success of matrix square-root normalized second-order pooling (SR-SOP) in deep CNN architectures [39, 27], we propose a parametric SR-SOP as CM of our GM-SOP.

**Parametric SR-SOP**    Given an input $\mathbf{X} \in \mathbb{R}^{L \times d}$ containing $L$ features of $d$-dimension, the SR-SOP of $\mathbf{X}$ is computed as

$$\mathbf{Z} = (\mathbf{X}^T \hat{\mathbf{J}} \mathbf{X})^{\frac{1}{2}} = \boldsymbol{\Sigma}^{\frac{1}{2}} = \mathbf{U}\boldsymbol{\Lambda}^{\frac{1}{2}}\mathbf{U}^T, \ \hat{\mathbf{J}} = \frac{1}{L}(\mathbf{I} - \frac{1}{L}\mathbf{1}\mathbf{1}^T), \tag{5}$$

where $\boldsymbol{\Sigma} = \mathbf{U}\boldsymbol{\Lambda}\mathbf{U}^T$ is eigenvalue decomposition (EIG) of $\boldsymbol{\Sigma}$. $\mathbf{I}$ and $\mathbf{1}$ are identity matrix and $L$-dimension vector with all elements being one, respectively. $\boldsymbol{\Sigma}$ is sample covariance (second-order statistics) of $\mathbf{X}$ estimated by the classical maximum likelihood estimation (MLE). Since $\mathbf{X}$ is a set of convolutional activations in deep CNNs, dimension of $\mathbf{X}$ is usually very high (128 in our case) while number of features is very small ($\sim 100$). It is well known that the classical MLE is not robust in the above scenario [10]. As explained in [40], performing matrix square root on covariance amounts to robust covariance estimation, very suitable for the scenario of high dimension and small sample. In addition, matrix square-root normalization can be regarded as a special case of Power-Euclidean metric [11] between covariance matrices, i.e., $\|\boldsymbol{\Sigma}_i^\beta - \boldsymbol{\Sigma}_j^\beta\|^2$ with $\beta = 0.5$, which is an approximation of Log-Euclidean metric [2] (hence making use of Riemannian geometry lying in covariance matrices[5]) while overcoming some downsides of Log-Euclidean metric [11].

Although SR-SOP in Eq. (5) benefits from some merits and achieves promising performance, it is a parameter-free model, which can not be trained as personalized CMs. Meanwhile, covariance $\boldsymbol{\Sigma}$ in Eq. (5) is calculated based on assumption that $\mathbf{X}$ is sampled from a Gaussian distribution, which may not always hold true. To handle above two problems, we propose a parametric second-order pooling (SOP), i.e.,

$$\boldsymbol{\Sigma}(\mathbf{Q}_j) = \mathbf{X}^T \mathbf{Q}_j \mathbf{X} = (\mathbf{P}_j \mathbf{X})^T (\mathbf{P}_j \mathbf{X}). \tag{6}$$

Different from the original sample covariance $\boldsymbol{\Sigma}$ with constant matrix $\hat{\mathbf{J}}$, $\mathbf{Q}_j$ in Eq. (6) is a learnable parameter, and $\mathbf{Q}_j$ is a symmetric positive semi-definite matrix with $\mathbf{Q}_j = \mathbf{P}_j^T \mathbf{P}_j$. Note that our parametric SOP in Eq. (6) shares similar philosophy with estimating covariance by assuming features follow a multivariate generalized Gaussian distribution with zero mean [32], i.e.,

$$p(\mathbf{x}_l; \widehat{\boldsymbol{\Sigma}}; \delta; \varepsilon) = \frac{\Gamma(d/2)}{\pi^{d/2}\Gamma(d/2\delta)2^{d/2\delta}} \frac{\delta}{\varepsilon^{d/2}|\widehat{\boldsymbol{\Sigma}}|^{1/2}} \exp\left( -\frac{1}{2\varepsilon^\delta}(\mathbf{x}_l \widehat{\boldsymbol{\Sigma}}^{-1} \mathbf{x}_l^T)^\delta \right), \tag{7}$$

where $\varepsilon$ and $\delta$ are parameters of scale and shape, respectively; $\widehat{\Sigma}$ is covariance matrix, and $\Gamma$ is a Gamma function. Compared with assumption of data distribution being a Gaussian in Eq. (5), generalized Gaussian distribution in Eq. (7) is more general and captures more complex characteristics. Given $\delta$ and $\varepsilon$, covariance matrix $\widehat{\Sigma}$ can be estimated using iterative reweighed methods [3, 43], and specifically, for the $j$-th iteration

$$\widehat{\Sigma}_j = \frac{1}{L} \sum_{l=1}^{L} \frac{Ld}{\mathbf{q}_l^j + (\mathbf{q}_l^j)^{1-\delta} \sum_{k \neq l} (\mathbf{q}_k^j)^{\delta}} \cdot \mathbf{x}_l^T \mathbf{x}_l = \frac{1}{L} \sum_{l=1}^{L} f_j(\mathbf{x}_l) \cdot \mathbf{x}_l^T \mathbf{x}_l = \mathbf{X}^T \widehat{\mathbf{G}}_j \mathbf{X}, \qquad (8)$$

where $\mathbf{q}_l^j = \mathbf{x}_l \widehat{\Sigma}_{j-1} \mathbf{x}_l^T$ and $\widehat{\mathbf{G}}_j$ is a diagonal matrix with diagonal elements being $\{f_j(\mathbf{x}_1)/L, \ldots, f_j(\mathbf{x}_L)/L\}$. It is worth mentioning at this point that our parametric SOP in Eq. (6) learns a more informative, full matrix, instead of only the diagonal one in traditional iterative reweighted methods [3, 43].

Obviously our parametric SOP in Eq. (6) can be regarded as single step of iterative estimation. To accomplish multi-step iterative estimation, we can learn a sequence of parameters $\mathbf{Q}_j$, $j = 1, \ldots, J$. After that, we perform matrix square-root normalization to obtain better performance. In practice we adopt two-step estimation (i.e., $J = 2$) to balance efficiency and effectiveness. We mention that implementation of each one of the two step estimation (i.e., $\mathbf{P}_j \mathbf{X}$) in Eq. (6) can be conveniently implemented using $1 \times 1$ convolution. As a result, our parametric SR-SOP can be transformed into learning multiple sequential $1 \times 1$ convolution operations following by computation of SR-SOP.

**Fast Iterative Algorithm** The computation of matrix square root in our parametric SR-SOP depends on EIG, which is limited supported on GPU, slowing down training speed of the whole network. Therefore, we employ the recently proposed iterative method [26] to speed up computing matrix square root. This method is based on Newton-Schulz iteration [14], which computes approximate matrix square root through iterative matrix multiplications as

$$\Sigma^{\frac{1}{2}} \approx \mathbf{A}_{\tilde{J}} : \{\mathbf{A}_{\tilde{j}} = \frac{1}{2}\mathbf{A}_{\tilde{j}-1}(3\mathbf{I} - \mathbf{B}_{\tilde{j}-1}\mathbf{A}_{\tilde{j}-1}); \mathbf{B}_{\tilde{j}} = \frac{1}{2}(3\mathbf{I} - \mathbf{B}_{\tilde{j}-1}\mathbf{A}_{\tilde{j}-1})\mathbf{B}_{\tilde{j}-1}\}_{\tilde{j}:=1,\cdots,\tilde{J}}, \quad (9)$$

where $\mathbf{A}_0 = \Sigma$ and $\mathbf{B}_0 = \mathbf{I}$. Clearly, Eq. (9) involves only matrix multiplications, more suitable for GPU implementation, and its back-propagation algorithm can be derived based on matrix back-propagation method [20]. Readers can refer to [26] for more details.

### 2.3 Deep CNN with GM-SOP

The overview of deep CNNs with our GM-SOP is illustrated in Figure 1. Notably, the proposed GM-SOP, rather than global average pooling or second-order pooling, is inserted after the last convolution layer. In our GM-SOP, the outputs of the last convolution layer are simultaneously fed into sparsity-constrained gating module and the bank of parametric CMs. In terms of the Top-$K$ results, the gating module allocates individual training samples to different CMs, and for each sample the final representation is a weighted sum of the outputs of $K$ selected CMs. We add a batch normalization [18] layer and a dropout [35] layer with drop rate of $0.2$ after final representation. Finally, we use a fully-connected layer and a *softmax* layer for classification. The sparsity-constrained gating module is composed of prediction layers, Top-$K$ and *softmax* operations, where the prediction layers share the same architecture with CMs to keep pace with representation. The parametric SR-SOP contains a set of convolution operations and iterative matrix multiplications. Clearly back-propagation of all involved layers can be accomplished according to traditional chain rule and matrix back-propagation method [20], and thus the deep CNNs with GM-SOP can be trained in an end-to-end manner.

## 3 Experiments

To evaluate the proposed method, we conduct experiments on two large-scale image benchmarks, i.e., downsampled ImageNet-1K [6] and Places365 [44]. We first describe implementation details of different competing methods, and then assess the effect of key parameters on our method using downsampled ImageNet-1K. Finally, we report the comparison results on two benchmarks.

Table 1: Modified ResNet-18 and ResNet-50 for downsampled ImageNet-1K and Places365.

| | conv1 | conv2_x | conv3_x | conv4_x | conv5_x | |
|---|---|---|---|---|---|---|
| ResNet-18 | $3 \times 3, 16$ (stride=1) | $\begin{bmatrix} 3 \times 3, 16 \\ 3 \times 3, 16 \end{bmatrix} \times 2$ | $\begin{bmatrix} 3 \times 3, 32 \\ 3 \times 3, 32 \end{bmatrix} \times 2$ | $\begin{bmatrix} 3 \times 3, 64 \\ 3 \times 3, 64 \end{bmatrix} \times 2$ | $\begin{bmatrix} 3 \times 3, 128 \\ 3 \times 3, 128 \end{bmatrix} \times 2$ | GAP |
| ResNet-50 | $3 \times 3, 16$ (stride=1) | $\begin{bmatrix} 3 \times 3, 16 \\ 3 \times 3, 16 \end{bmatrix} \times 6$ | $\begin{bmatrix} 3 \times 3, 32 \\ 3 \times 3, 32 \end{bmatrix} \times 6$ | $\begin{bmatrix} 3 \times 3, 64 \\ 3 \times 3, 64 \end{bmatrix} \times 6$ | $\begin{bmatrix} 3 \times 3, 128 \\ 3 \times 3, 128 \end{bmatrix} \times 6$ | GAP |
| Output Size | $64 \times 64$ $96 \times 96$ | $64 \times 64$ $96 \times 96$ | $32 \times 32$ $48 \times 48$ | $16 \times 16$ $24 \times 24$ | $8 \times 8$ $12 \times 12$ | ImageNet-1K Places365 |

## 3.1 Implementation Details

In this work, we implement several methods for comparison, and consider two basic CNN models including ResNet [13] of 18 and 50 layers. All competing methods are described as follows.

(1) *ResNet-18/ResNet-50* indicate original ResNets with first-order GAP.

(2) *ResNet-18-Xd/ResNet-50-Xd* denote ResNets with a parametric GAP, which is achieved by inserting a $1 \times 1 \times X$ convolution layer before GAP. Such method can be regarded as a special case of gated mixture of first-order GAP with single CM.

(3) *Ave-GAP-K* performs simple average of $K$ parametric GAPs without gating module.

(4) *GM-GAP-N-K* selects $K$ parametric GAPs from $N$ GAP candidates through sparsity-constrained gating module, and performs weighted sum of $K$ selected parametric GAPs.

(5) *Parametric SR-SOP* is achieved by adding two convolution layers of $\{1 \times 1 \times 128 \times 256\}$ and $\{1 \times 1 \times 256 \times 128\}$ before SR-SOP.

(6) *Ave-SOP-K* performs simple average of $K$ parametric SR-SOPs without gating module.

(7) *GM-SOP-N-K* selects $K$ parametric SR-SOP models from $N$ candidates with sparsity-constrained gating module, and performs weighted sum of $K$ selected candidates.

In our experiments, image sizes of downsampled ImageNet-1K and Places365 respectively are $64 \times 64$ and $100 \times 100$, so we modify ResNet architectures in [13] to fit image sizes in our case. The architectures of modified ResNet-18 and ResNet-50 are given in Table 1. As suggested in [26], we compute approximate matrix square root in Eq. (9) within five iterations to balance the effectiveness and efficiency. For training the whole network, we employ mini-batch stochastic gradient descent with batchsize of 256 and momentum of 0.9. The parameter of weight decay is set to 5e-4. The program is implemented using MatConvNet toolkit [37], and runs on a PC equipped with an Intel i7-4790K@4.00GHz CPU, a single NVIDIA GeForce GTX1080 GPU and 64G RAM.

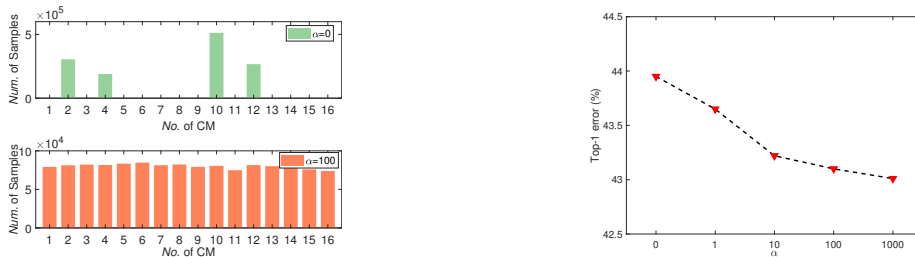

Figure 2: Left: Numbers of receiving samples in each CM by setting $\alpha = 0$ and $\alpha = 100$. $\alpha = 0$ indicates balance loss is not employed. Right: Results of GM-GAP-16-4 with various $\alpha$.

## 3.2 Ablation Studies on Downsampled ImageNet-1K

Our gated mixture model has three key parameters, i.e., weight parameter $\alpha$ of balance loss in Eq. (4), number of CMs ($N$) and number of selected CMs ($K$). We evaluate them using gated mixture of first-order GAP with ResNet-18 on downsampled ImageNet-1K, and the decided optimal parameters are directly adopted to GM-SOP. Such strategy is not only faster but also avoids over-fitting on parameters of GM-SOP. We train the networks on downsampled ImageNet-1K dataset [6] within

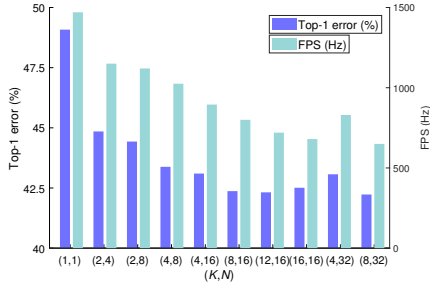

Figure 3: GM-GAP with various $N$ and $K$.

| Methods | Dim. of Reps. | Top-1 error (%) |
|---|---|---|
| ResNet-18 | 128 | 52.00 |
| ResNet-18-512d | 512 | 49.08 |
| Ave-GAP-16 | 512 | 47.44 |
| GM-GAP-16-8 (Ours) | 512 | **42.37** |
| ResNet-18-8256d | 8256 | 47.29 |
| SR-SOP | 8256 | 40.32 |
| Ave-SOP-16 | 8256 | 40.28 |
| Parametric SR-SOP | 8256 | 40.01 |
| GM-SOP-16-8 (Ours) | 8256 | **38.21** |

Table 2: Comparison with counterparts using ResNet-18 on downsampled ImageNet-1K.

Table 3: Comparison with state-of-the-arts on downsampled ImageNet-1K. The methods marked by $\star$ double number of training images using original images and their horizontal flipping ones, and perform 4 pixels padding and random crop in both training and prediction stages.

| Methods | Number of Parameters | Dimension of Representations | Top-1 error/Top-5 error (%) |
|---|---|---|---|
| WRN-36-1$^\star$ [6] | 1.6M | 128 | 49.79/24.17 |
| WRN-36-2$^\star$ [6] | 6.2M | 256 | 39.55/16.57 |
| WRN-36-5$^\star$ [6] | 37.6M | 640 | 32.34/12.64 |
| ResNet-18-512d [13] | 1.3M | 512 | 49.08/24.25 |
| ResNet-18+One-layer-SG-MOE [33] | 2.3M | 512 | 46.80/22.63 |
| ResNet-18+NetVLAD [1] | 8.9M | 8192 | 45.16/21.73 |
| ResNet-50 [13] | 2.4M | 128 | 43.28/19.39 |
| ResNet-50-512d [13] | 2.8M | 512 | 41.79/18.30 |
| ResNet-50-8256d [13] | 11.6M | 8256 | 41.42/18.14 |
| GM-GAP-16-8 + ResNet-18 (Ours) | 2.3M | 512 | **42.37/18.82** |
| GM-GAP-16-8 + ResNet-18$^\star$ (Ours) | 2.3M | 512 | **40.03/17.91** |
| GM-GAP-16-8 + WRN-36-2 (Ours) | 8.7M | 512 | **35.97/14.41** |
| GM-SOP-16-8 + ResNet-18 (Ours) | 10.3M | 8256 | **38.21/17.01** |
| GM-SOP-16-8 + ResNet-50 (Ours) | 11.9M | 8256 | **35.73/14.96** |
| GM-SOP-16-8 + WRN-36-2 (Ours) | 15.7M | 8256 | **32.33/12.35** |

$\{50, 15, 10\}$ epochs, while the initial learning rate is set to $0.075$ with decay rate of $0.1$. Only random flipping is used for data augmentation, and prediction is performed on whole images. Following the common settings in [13, 6], we run experiments three trials and report Top-1 error of different methods on validation set for comparison.

*Effect of Parameter $\alpha$* The goal of balance loss is to make as many CMs as possible be adequately trained. Here we evaluate its effect using GM-GAP-16-4 with various $\alpha$. Figure 2 (Left) compares numbers of receiving samples in each CM by setting $\alpha = 0$ and $\alpha = 100$ within the last training epoch, where $\alpha = 0$ indicates balance loss is discarded. Clearly, only four CMs receive almost all training samples when balance loss is not employed (i.e., $\alpha = 0$). Differently, the balance loss with $\alpha = 100$ makes most of CMs receive similar amount of training samples. Figure 2 (Right) shows the results of GM-GAP-16-4 with various $\alpha$, from it we can see that adequately training as many CMs as possible achieves lower classification error. The balance loss with $\alpha = 100, 1000$ obtain similar results. Without loss of generality, we set parameter $\alpha$ to 100 in following experiments.

*Numbers of $N$ and $K$* Then we assess the effect of numbers of $N$ and $K$ by setting $\alpha = 100$. Top-1 error and training speed (Frames Per Second, FPS) of GM-GAP with various $N$ and $K$ are illustrated in Figure 3. Fixing number of $K$, increase of $N$ leads lower error while bringing more computational costs. When number of $N$ is fixed, better results are obtained by appropriately enlarging $K$. Taking $N = 16$ as an example, $K = 8$ gets the best result, and results of $K = 12$ and $K = 16$ are slightly inferior to the one of $K = 8$. It maybe owe to the fact that sparsity constraint eliminates noisy CMs having small weights. In addition, we can see that GM-GAP with $N = 16, K = 8$ ($42.37\%, 800$Hz) employing 16 CMs is only about $1.8$ times slower than baseline ($49.08\%, 1470$Hz) with one CM, but achieves about $6.71\%$ gains. We also experiment with more CMs. GM-GAP with $N$=128 and $K$=32 obtains $42.52\%$, achieving no gain over the result of $N$=16 and $K$=8 ($42.37\%$). We observe larger number (128) of CMs leads to a bit over-fitting in our case and more computation cost. To balance efficiency and effectiveness, we set $N = 16$ and $K = 8$ for both GM-GAP and GM-SOP throughout all remaining experiments.

Table 4: Comparison with counterparts on Places365 dataset with image size of $100 \times 100$.

| | ResNet-18-512d | GM-GAP-16-8 | ResNet-18-8256d | SR-SOP | Parametric SR-SOP | GM-SOP-16-8 |
|---|---|---|---|---|---|---|
| Dim. | 512 | 512 | 8256 | 8256 | 8256 | 8256 |
| Top-1 error (%) | 49.96 | **48.07** | 49.99 | 48.11 | 47.48 | **47.18** |
| Top-5 error (%) | 19.19 | **17.84** | 19.32 | 18.01 | 17.52 | **17.02** |

*Comparison with Counterparts* We compare our method with several counterparts, and results of different methods are listed in Table 2. We train SR-SOP (or parametric SR-SOP) and Ave-SOP-16 (or GM-SOP-16-8) within $\{20, 5, 5, 5\}$ and $\{40, 10, 5, 5\}$ epochs. The initial learning rates are set to 0.15 and 0.1 with decay rate of 0.1. When employing GAP as CM, our GM-GAP is superior to ResNet-18-512d (single CM) and Ave-GAP-16 (direct ensemble) by a large margin. Meanwhile, SR-SOP performs better than GM-GAP, and improves ResNet-18-8256d by $6.97\%$ with the same dimensional representation, demonstrating superiority of SOP. Note that our parametric SR-SOP outperforms original SR-SOP with negligible additional costs (680Hz $vs.$ 670Hz), and they are moderately slower than GM-GAP-16-8 (800Hz). The GM-SOP-16-8 outperforms Ave-SOP-16 by $2.07\%$ with more than 2 times faster, and improves SR-SOP by about $2.11\%$ with about 2 times slower. The above results verify the effectiveness of our GM-SOP and idea of gated mixture model.

### 3.3 Results on Downsampled ImageNet-1K

Here we compare our method with state-of-the-art (SOTA) methods on downsampled ImageNet-1K [6]. Since this dataset is recently proposed and has few reported results, we implement several SOTA methods based on the modified ResNet-18 and ResNet-50 by ourselves and report their results with trying our best to tune their hyper-parameters. NetVLAD [1] is implemented using public available source code with setting dictionary size to 64. By using the same settings with GM-GAP, we replace GAP with One-layer-SG-MoE [33], where each expert is a $128 \times 512$ fully-connected layer. All ResNet-50 based methods are trained within $\{50, 15, 10\}$ epochs, and initial learning rates with decay rate of 0.1 are set to 0.1 and 0.075 for our GM-SOP and remaining ones, respectively. We also compare with wide residual network (WRN) [42], whose results are duplicated from [6]. As shown in Table 3, our GM-SOP and GM-GAP significantly outperform NetVLAD, One-layer-SG-MoE and original network, when ResNet-18 is employed. Meanwhile, our GM-SOP with ResNet-50 improves original network and its variants by a large margin. These results verify our methods effectively improve existing deep CNNs. Our GM-SOP with ResNet-18 clearly outperforms WRN-36-1 and WRN-36-2, although the latter ones adopt more sophisticated data augmentation. By using the same augmentation strategy in [6], GM-GAP-16-8 achieves over $2\%$ gains in Top-1 error, which uses much less parameters to get simliar results with WRN-36-2. To further evaluate our methods, we integrate the proposed methods with the stronger WRN-36-2, our GM-GAP and GM-GOP improve WRN-36-2 over $3.58\%$ and $7.22\%$ in Top-1 error, respectively. Note that GM-SOP with WRN-36-2 obtains the similar result with WRN-36-5 [6] using one half parameters.

### 3.4 Results on Places365

Finally, we evaluate our method on Places365 [44], which contains about 1.8 million training images and 36,500 validation images collected from 365 scene categories. In our experiments, we resize all images to $100 \times 100$, developing a downsampled Places365 dataset. It is much larger and more challenging than existing low-resolution image datasets [24, 6]. We implement several counterparts and compare with our method based on ResNet-18. For training these networks, we randomly crop a $96 \times 96$ image patch or its flip as input. ResNet-18-8256d and remaining ones are trained within $\{35, 10, 10, 5\}$ and $\{25, 5, 5, 5\}$ epochs, and the initial learning rates are set to 0.05 and 0.1 with decay rate of 0.1, respectively. The inference is performed on single center crop, and we report results on validation set for comparison. The results of different methods are given in Table 4, from it we can see that our GM-SOP-16-8 achieves the best result and significantly outperforms ResNet-18-512d and ResNet-18-8256d, further demonstrating the effectiveness of GM-SOP. Meanwhile, GM-GAP-16-8 and GM-SOP-16-8 are superior to ResNet-18-512d and SR-SOP by a large margin, respectively. It indicates the idea of gated mixture model is helpful for improving representation ability of deep CNNs. Note that parametric SR-SOP for non-trivial gains over original SR-SOP, showing a more general approach for image modeling is meaningful and useful for improving performance.

# 4 Related Work

Our GM-SOP method shares similarity with sparsely-gated mixture-of-experts (SG-MoE) layer [33]. The SG-MoE motivates the gating module of our GM-SOP, but quite differently, our GM-SOP proposes a parametric SR-SOP as CM while SG-MoE employs a linear transformation (fully-connected layer) as expert. Meanwhile, our methods significantly outperform one SG-MoE layer. Additionally, the SG-MoE is proposed as a general purpose component in a recurrent model [15], while our GM-SOP is proposed as a global modeling step to improve representation ability of deep CNNs. This work also is related to those methods integrating single HOP into deep CNNs [19, 29, 39, 27, 8]. Beyond them, our GM-SOP is a mixture model, which can capture richer information and achieve better performance. NetVLAD [1] and MFAFVNet [28] extend deep CNNs with popular feature encoding methods, which also can be seen as mixture models. However, different from their concatenation scheme for all CMs, our GM-SOP performs sum of selected CMs, leading more compact representations. Meanwhile, our GM-SOP is clearly superior to feature encoding based NetVLAD [1]. Recently, some researchers propose to learn deep mixture probability models for semi-supervised learning [31] and unsupervised clustering [38]. These methods formulate mixture probability models as multi-layer networks, and infer the corresponding networks with deriving variants of EM algorithm. In contrary to deep mixture probability models [31, 38], we aim at plugging a trainable gated mixture model into deep CNNs as representation for supervised classification.

# 5 Conclusion

This paper proposes a novel GM-SOP method for improving deep CNNs, whose core is a trainable gated mixture of parametric second-order pooling model for summarizing the outputs of the last convolution layer as image representation. The GM-SOP can be flexibly integrated into deep CNNs in an end-to-end manner. Compared with popular GAP and existing HOP methods only considering unimodal distributions, our GM-SOP can make better use of statistical information inherent in convolutional activations, leading better representation ability and higher accuracy. The experimental results on two large-scale image benchmarks demonstrate the gated mixture model is helpful to improve classification performance of deep CNNs, and our GM-SOP method clearly outperforms its counterparts with affordable costs. Note that the proposed GM-SOP is an architecture-independent model, so we can flexibly adopt it to other advanced CNN architectures [17, 5, 16]. In future, we will experiment with standard-size ImageNet dataset, and extend GM-SOP to other tasks, such as video classification and semantic segmentation.

### Acknowledgments

The work was supported by the National Natural Science Foundation of China (Grant No. 61471082, 61671182, 61806140) and the State Key Program of National Natural Science Foundation of China (Grant No. 61732011). Qilong Wang was supported by China Post-doctoral Programme Foundation for Innovative Talent. We thank NVIDIA corporation for donating GPU.

## Footnotes

[4]We reasonably introduce a group of trainable parameters into SR-SOP, and use the recently proposed fast iterative algorithm [26] to speed up matrix square-root normalization on GPU.

[5]Covariance matrices are symmetric positive definite matrices, whose space forms a non-linear Riemannian manifold [2].

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
