[Supplementary Material · supplementary.pdf]

# Supplementary Material for Global Gated Mixture of Second-order Pooling for Improving Deep Convolutional Neural Networks

**Qilong Wang**[1,2,*,†], **Zilin Gao**[2,*], **Jiangtao Xie**[2], **Wangmeng Zuo**[3], **Peihua Li**[2,‡]

[1]Tianjin University, [2]Dalian University of Technology, [3] Harbin Institute of Technology
qlwang@tju.edu.cn, gzl@mail.dlut.edu.cn, jiangtaoxie@mail.dlut.edu.cn
wmzuo@hit.edu.cn, peihuali@dlut.edu.cn

## 1 Relationship between Parametric SOP and Covariance of Multivariate Generalized Gaussian Distribution

Here, we show our parametric second-order pooling (SOP) shares similar philosophy with estimation of covariance by assuming features are sampled from a generalized multivariate Gaussian distribution with zero mean. Firstly, our parametric SOP takes the following form:

$$\mathbf{\Sigma}(\mathbf{Q}_j) = \mathbf{X}^T \mathbf{Q}_j \mathbf{X} = (\mathbf{P}_j \mathbf{X})^T (\mathbf{P}_j \mathbf{X}), \tag{1}$$

where $\mathbf{Q}_j$ is a learnable matrix, and $\mathbf{Q}_j$ is a symmetric positive definite matrix, which has a unique decomposition $\mathbf{Q}_j = \mathbf{P}_j^T \mathbf{P}_j$. Given a set of $\mathbf{X} \in \mathbb{R}^{L \times d} = \{\mathbf{x}_1, \ldots, \mathbf{x}_L\}$, their generalized multivariate Gaussian distribution with zero mean [5] can be represented as

$$p(\mathbf{x}_l; \widehat{\mathbf{\Sigma}}; \delta; \varepsilon) = \frac{\Gamma(d/2)}{\pi^{d/2}\Gamma(d/2\delta)2^{d/2\delta}} \frac{\delta}{\varepsilon^{d/2}|\widehat{\mathbf{\Sigma}}|^{1/2}} \exp\left(-\frac{1}{2\varepsilon^\delta}(\mathbf{x}_l \widehat{\mathbf{\Sigma}}^{-1} \mathbf{x}_l^T)^\delta\right), \tag{2}$$

where $\varepsilon$ and $\delta$ are parameters of scale and shape, respectively; $\widehat{\mathbf{\Sigma}}$ is covariance matrix, and $\Gamma$ is a Gamma function. Under maximum likelihood criterion, given $\delta$ and $\varepsilon$, covariance matrix $\widehat{\mathbf{\Sigma}}$ can be estimated by:

$$\arg\min_{\widehat{\mathbf{\Sigma}}} \sum_{l=1}^{L} (\mathbf{x}_l \widehat{\mathbf{\Sigma}}^{-1} \mathbf{x}_l^T)^\delta + N \log|\widehat{\mathbf{\Sigma}}|. \tag{3}$$

As shown in [1, 6], the objective function in Eq. (3) can converge to a stationary point by using iterative reweighed methods, whose $j$-th iteration has the following form:

$$\widehat{\mathbf{\Sigma}}_j = \frac{1}{L} \sum_{l=1}^{L} \frac{Ld}{\mathbf{q}_l^j + (\mathbf{q}_l^j)^{1-\delta} \sum_{k \neq j} (\mathbf{q}_k^j)^\delta} \cdot \mathbf{x}_l^T \mathbf{x}_l, \quad \mathbf{q}_l^j = \mathbf{x}_l \widehat{\mathbf{\Sigma}}_{j-1} \mathbf{x}_l^T. \tag{4}$$

Let $f_j(\mathbf{x}_l) = \frac{Ld}{\mathbf{q}_l^j + (\mathbf{q}_l^j)^{1-\delta} \sum_{k \neq l} (\mathbf{q}_k^j)^\delta}$, we have

$$\widehat{\mathbf{\Sigma}}_j = \mathbf{X}^T \widehat{\mathbf{G}}_j \mathbf{X} = (\widehat{\mathbf{R}}_j \mathbf{X})^T (\widehat{\mathbf{R}}_j \mathbf{X}), \tag{5}$$

where $\widehat{\mathbf{G}}_j$ and $\widehat{\mathbf{R}}_j$ are diagonal matrices, and their diagonal elements are $\{f_j(\mathbf{x}_1)/L, \ldots, f_j(\mathbf{x}_L)/L\}$ and $\{\sqrt{f_j(\mathbf{x}_1)/L}, \ldots, \sqrt{f_j(\mathbf{x}_L)/L}\}$, respectively. Comparing Eq. (1) with Eq. (5), it is evident that,

in each iteration, our parametric SOP learns a full matrix $\mathbf{P}_j$, while iterative reweighted methods [1, 6] learn the diagonal $\widehat{\mathbf{R}}_j$.

According to Eq. (5), iterative reweighted methods can be accomplished by $J$ iterations:

$$\widehat{\boldsymbol{\Sigma}} = (\widehat{\mathbf{R}}_T \cdots \widehat{\mathbf{R}}_1 \mathbf{X})^T (\widehat{\mathbf{R}}_T \cdots \widehat{\mathbf{R}}_1 \mathbf{X}), \tag{6}$$

Correspondingly we can learn a sequence of parameters $\mathbf{Q}_j$, $\{j = 1, \ldots, J\}$ for our parametric SOP, i.e.,

$$\boldsymbol{\Sigma} = (\mathbf{P}_T \cdots \mathbf{P}_1 \mathbf{X})^T (\mathbf{P}_T \cdots \mathbf{P}_1 \mathbf{X}). \tag{7}$$

Since $\mathbf{P}_j \mathbf{X}$ can be conveniently implemented using $1 \times 1$ convolution, our parametric SOP can be transformed into learning multiple sequential $1 \times 1$ convolution operations following by computation of SOP. Eqs. (5) and (7) clearly show our parametric SOP and covariance of multivariate generalized Gaussian distribution share the similar form.

## 2 Details of Matrix Square Root of Covariance Based on Newton-Schulz Iteration [2]

Let $\mathbf{A}_0 = \boldsymbol{\Sigma}$ and $\mathbf{B}_0 = \mathbf{I}$, according to Newton-Schulz iteration [2], we have

$$\mathbf{A}_{\tilde{j}} = \frac{1}{2} \mathbf{A}_{\tilde{j}-1}(3\mathbf{I} - \mathbf{B}_{\tilde{j}-1}\mathbf{A}_{\tilde{j}-1}); \ \ \mathbf{B}_{\tilde{j}} = \frac{1}{2}(3\mathbf{I} - \mathbf{B}_{\tilde{j}-1}\mathbf{A}_{\tilde{j}-1})\mathbf{B}_{\tilde{j}-1}, \tag{8}$$

where $\mathbf{A}_{\tilde{j}}$ and $\mathbf{B}_{\tilde{j}}$ will converge to $\boldsymbol{\Sigma}^{\frac{1}{2}}$ and $\boldsymbol{\Sigma}^{-\frac{1}{2}}$ after $\tilde{J}$ iterations, respectively. However, Eq. (8) requires norm of $(\mathbf{I} - \boldsymbol{\Sigma})$, i.e., $\|\mathbf{I} - \boldsymbol{\Sigma}\| < 1$. The recently proposed method [4] introduces pre-normalization (i.e., $\tilde{\boldsymbol{\Sigma}} = \frac{1}{\mathrm{tr}(\boldsymbol{\Sigma})}\boldsymbol{\Sigma}$) and post-compensation operations (i.e., $\mathbf{Z} = \sqrt{\mathrm{tr}(\boldsymbol{\Sigma})}\mathbf{A}_{\tilde{j}}$) for Newton-Schulz iteration in Eq. (8), and develop a back-propagation (BP) algorithm based on matrix back-propagation method [3] for end-to-end learning. Specifically, given the loss function $l$, BP for post-compensation can be achieved by

$$\frac{\partial l}{\partial \mathbf{A}_{\tilde{j}}} = \sqrt{\mathrm{tr}(\boldsymbol{\Sigma})}\frac{\partial l}{\partial \mathbf{Z}}; \ \ \frac{\partial l}{\partial \boldsymbol{\Sigma}}\Big|_{\mathrm{post}} = \frac{1}{2\sqrt{\mathrm{tr}(\boldsymbol{\Sigma})}}\mathrm{tr}\Big(\Big(\frac{\partial l}{\partial \mathbf{Z}}\Big)^T \mathbf{A}_{\tilde{j}}\Big)\mathbf{I}. \tag{9}$$

Let $\frac{\partial l}{\partial \mathbf{B}_{\tilde{j}}} = 0$, for $\tilde{j} = \tilde{J}, \ldots, 2$, BP of Newton-Schulz iteration can be accomplished with

$$\frac{\partial l}{\partial \mathbf{A}_{\tilde{j}-1}} = \frac{1}{2}\Big(\frac{\partial l}{\partial \mathbf{A}_{\tilde{j}}}\Big(3\mathbf{I} - \mathbf{A}_{\tilde{j}-1}\mathbf{B}_{\tilde{j}-1}\Big) - \mathbf{B}_{\tilde{j}-1}\frac{\partial l}{\partial \mathbf{B}_{\tilde{j}}}\mathbf{B}_{\tilde{j}-1} - \mathbf{B}_{\tilde{j}-1}\mathbf{A}_{\tilde{j}-1}\frac{\partial l}{\partial \mathbf{A}_{\tilde{j}}}\Big)$$

$$\frac{\partial l}{\partial \mathbf{B}_{\tilde{j}-1}} = \frac{1}{2}\Big(\Big(3\mathbf{I} - \mathbf{A}_{\tilde{j}-1}\mathbf{B}_{\tilde{j}-1}\Big)\frac{\partial l}{\partial \mathbf{B}_{\tilde{j}}} - \mathbf{A}_{\tilde{j}-1}\frac{\partial l}{\partial \mathbf{A}_{\tilde{j}}}\mathbf{A}_{\tilde{j}-1} - \frac{\partial l}{\partial \mathbf{B}_{\tilde{j}}}\mathbf{B}_{\tilde{j}-1}\mathbf{A}_{\tilde{j}-1}\Big). \tag{10}$$

When $\tilde{j} = 1$, we have

$$\frac{\partial l}{\partial \tilde{\boldsymbol{\Sigma}}} = \frac{1}{2}\Big(\frac{\partial l}{\partial \mathbf{A}_1}\Big(3\mathbf{I} - \tilde{\boldsymbol{\Sigma}}\Big) - \frac{\partial l}{\partial \mathbf{B}_1} - \tilde{\boldsymbol{\Sigma}}\frac{\partial l}{\partial \mathbf{A}_1}\Big). \tag{11}$$

Finally, BP of pre-normalization can be computed as

$$\frac{\partial l}{\partial \boldsymbol{\Sigma}} = -\frac{1}{(\mathrm{tr}(\boldsymbol{\Sigma}))^2}\mathrm{tr}\Big(\Big(\frac{\partial l}{\partial \tilde{\boldsymbol{\Sigma}}}\Big)^T \boldsymbol{\Sigma}\Big)\mathbf{I} + \frac{1}{\mathrm{tr}(\boldsymbol{\Sigma})}\frac{\partial l}{\partial \tilde{\boldsymbol{\Sigma}}} + \frac{\partial l}{\partial \boldsymbol{\Sigma}}\Big|_{\mathrm{post}}. \tag{12}$$

Eq. (12) is the gradient of loss function $l$ with respect to $\boldsymbol{\Sigma}$, which is used to achieve BP for matrix square root of covariance. Readers can refer to [4] for more details.

## Footnotes

*The first two authors contribute equally to this work.

†This work was mainly done when he was with Dalian University of Technology.

‡Peihua Li is the corresponding author.