[Reviews · NeurIPS 2018]

Reviewer 1



The idea behind a Sparse Gated Mixture (GM) of Expert model has already been proposed in [1]. The main novelty of this paper is in the way the sparse mixture model is applied, namely to modify the 2nd order pooling layer within a deep CNN model to have a bank of candidates. The way GM works is as follow: Given an input sample, the sparsity-constrained gating module adaptively selects Top-K experts from N candidates according to assigned weights and outputs the weighted sum of the outputs of the K selected experts. Another contribution of this paper is to define a parameterized architecture for pooling: For the choice of expert, the authors use a modified learnable version of matrix square-root normalized second-order pooling (SR-SOP) [2] . --------------- The idea of using sparsely gated experts is very promising and finding new ways to apply it within model architectures, as it is the case in this work, is important. The experiments first show that SR-SOP is advantageous over regular SOP, both are prior work, but it's good to justify why use SR-SOP in the first place. It then shows results on the new parameterized version (which slightly improves results) and then results on adding the GM which makes 2+% overall improvements. While the presented results show advantage of this method, the experiments could be stronger and more convincing. A down-sampled ImageNet is used as dataset and a modified ResNet is used for baseline. Why not experimenting with the standard-size ImageNet, or CIFAR10-100 datasets (especially, if choosing to go with a smaller size image)? Much more research has been done on these 2 datasets and improving over those strong baselines would be a much more convincing result. Have the authors tried increasing the number of CMs (component modules, aka experts) to multiple hundreds or thousands, to see the limit of adding capacity to the model? The maximum # of CMs in this paper is 32 and it looks like the best overall result is achieved with that many CMs. In a prior work [1], some of the best results are achieved by having 1-2k experts. [1] N. Shazeer, A. Mirhoseini, K. Maziarz, A. Davis, Q. V. Le, G. E. Hinton, and J. Dean. Outrageously large neural networks: The sparsely-gated mixture-of-experts layer. In ICLR, 2017. [2] P. Li, J. Xie, Q. Wang, and W. Zuo. Is second-order information helpful for large-scale visual recognition? In ICCV, 2017

Reviewer 2



This work focuses on the representation of the pooling layer in CNN. Using CNNs, the pooling layer usually run the maximum pooling or average pooling, also called the first order pooling. To improve the representation of pooling layer, this paper presents the second order representation in statistics which includes mean and variance. Finally, the authors impose the sparsity constraint to estimate the weights for different second order representations. Pros : 1. Typical CNN usually extracts first-order information in pooling layer. This work extends the extraction of second-order information to represent the pooling layer, such as Gaussian distribution. Complete representation is helpful for prediction. 2. To reduce the information loss or the instability caused by dimension reduction in pooling layers, this work adopts the matrix decomposition and uses the parametric method to carry out the square-root normalized second-order representation. Cons : 1. The idea of the mixture model is the key contribution in this paper, but the weight of each component models seems to define by experience. 2. Although the performance of this idea is very good, there’s no novel mathematical proved in this paper.

Reviewer 3



Global Averaging Pooling (GAP) and Higher-order Pooling (HOP) assume a unimodal distribution, which cannot adequately capture statistics of convolutional activations. This paper proposes a Gated Mixture of Second-order Pooling (GM-SOP) to improve the representation capacity of deep CNN. GM-SOP represents the feature distribution by a weighted sum of component pooling models, and weights are sparsely determined by Sparsity-constrained Gating Module [32]. For the component model, this paper proposes a parametric version of square-root normalized second-order pooling (SR-SOP)[26] which inserts two 1x1 sequential convolution layers before SR-SOP, which can be viewed as an iterative reweighted estimation of covariance matrix of a generalized multivariate Gaussian distribution. The experiments on the downsampled ImageNet-1K and Places365 showed the effectiveness of the proposed method. Strengths: + The ideas to improve the second-order pooling by mixture models with gating functions seems to be reasonable. + Extensive experiments which justify each component of the proposed method. + The performance is better than other methods with the same dimension. + This paper is well written. Weaknesses: -The proposed idea is a bit incremental, which is a combination of sparsity-constrained Gated Module[32] and SR-SOP[26]. Comments: The parametric SR-SOP is justified by the iterative reweighted estimation of the covariance matrix of generalized multivariate Gaussian distribution with zero mean[42]. However, I could not find Eq.(7) in [42]. I would like to know if this citation is correct. The experiments of parameter analysis (\alpha, N, K) were conducted using the GAP as the component model. I wonder if the same trend can be seen when the proposed parametric SR-SOP is used for the component model. Rebuttal response: The rebuttal addressed the concerns about parameter analysis and the Eq.(7). I expect that the authors will add the mathematical details of the relationship between the proposed parametric SR-SOP and iterative estimation of generalized Gaussian distribution in the supplemental material.